# UK medical students' self-reported knowledge and harm assessment of psychedelics and their application in clinical research: a cross-sectional study

Charlie Song-Smith,[1] Edward Jacobs,[2,3] James Rucker  ,[4,5] Matthew Saint,[6] James Cooke,[7] Marco Schlosser  [8,9]

For numbered affiliations see end of article.

**Correspondence to**
Dr Marco Schlosser;
marco.schlosser@ucl.ac.uk

## ABSTRACT

**Objective** To capture UK medical students' self-reported knowledge and harm assessment of psychedelics and to explore the factors associated with support for changing the legal status of psychedelics to facilitate further clinical research.

**Design** Cross-sectional, anonymous online survey of UK medical students using a non-random sampling method.

**Setting** UK medical schools recognised by the General Medical Council.

**Participants** 132 medical students who had spent an average of 3.8 years (SD=1.4; range: 1–6) in medical school.

**Results** Most students (83%) reported that they were aware of psychedelic research and only four participants (3%) said that they were not interested in learning more about this type of research. Although medical students' harm assessment of psychedelics closely aligned with that of experts, only 17% of students felt well-educated on psychedelic research. Teachings on psychedelics were only rarely encountered in their curriculum (psilocybin: 14.1 (SD=19.9), scale: 0 (never) to 100 (very often)). Time spent at medical schools was not associated with more knowledge about psychedelics (r=0.12, p=0.129). On average, this sample of medical students showed strong support for changing the legal status of psychedelics to facilitate further research into their potential clinical applications (psilocybin: 80.2 (SD=24.8), scale: 0 (strongly oppose) to 100 (strongly support)). Regression modelling indicated that greater knowledge of psychedelics (p<0.001), lower estimated harm scores (p<0.001), more time spent in medical school (p=0.024) and lower perceived effectiveness of non-pharmacological mental health treatments (p=0.044) were associated with greater support for legal status change.

**Conclusions** Our findings reveal a significant interest among UK medical students to learn more about psychedelic research and a strong support for further psychedelic research. Future studies are needed to examine how medical education could be refined to adequately prepare medical students for a changing healthcare landscape in which psychedelic-assisted therapy could soon be implemented in clinical practice.

## INTRODUCTION

Since the early 1990s, there has been a steady resurgence of research involving psychedelics such as lysergic acid diethylamide (LSD),

## STRENGTHS AND LIMITATIONS OF THIS STUDY

⇒ This study presents the first survey of UK medical students' attitudes towards psychedelics and their application in clinical research.
⇒ The wide-ranging assessment included medical students' self-reported knowledge of psychedelics, a multidimensional harm assessment of psychedelics, level of support for changing the legal status of psychedelics to facilitate further research into their potential clinical applications, and students' perceptions of medical education on psychedelics and psychedelic research.
⇒ The cross-sectional nature of our data prevented us from drawing causal conclusions about potential changes in students' attitudes and knowledge over time.
⇒ The limited range of demographic variables did not allow for an in-depth characterisation of this sample and important predictors of attitudes might have been missed. Non-random sampling and selection bias could have impacted the generalisability of our findings as students already interested in drugs might have been more likely to participate.

psilocybin and N, N-Dimethyltryptamine.[1–3] After a short-lived period of medical use of psychedelics by clinicians and researchers in the middle of the 20th century, psychedelic research was effectively ended by prohibitive legislation and a lack of governmental research funding.[4] The clinical trials conducted prior to these legal sanctions suggested that psychedelics—which were frequently used in combination with or as an aid to psychotherapy—could positively impact a range of disorders, particularly mood disorders and alcohol dependence.[5 6] These early findings have been corroborated by modern-era clinical trials, which support the therapeutic effects of psychedelics and psychedelic-assisted psychotherapy in some patient groups.[7–10] The largest published and

ongoing psychedelic trials investigate the effects of psilocybin in combination with psychological support for the treatment of major depressive disorder.[11–15] Clinical investigations into the therapeutic efficacy of related drugs, such as 3,4-methylenedioxymethamphetamine (MDMA) for post-traumatic stress disorder (PTSD)[16 17] and ketamine for depression,[18–20] have also advanced substantially. In the context of the present study, the term 'psychedelic' will encompass the classic psychedelics (e.g., psilocybin, LSD) as well as MDMA and ketamine.

If current developments continue unhindered, some psychedelics and forms of psychedelic-assisted therapy will likely receive regulatory approval and thus transition from legally restricted use in research studies to integration in clinical practice.[21] Several companies are undertaking efforts to commercialise psychedelics as medical treatments.[22] The extent to which psychedelics will be clinically implemented and disseminated will also depend on medical professionals' knowledge and attitudes towards these drugs. Despite the looming possibility of regulatory approval, research on medical professionals' opinions surrounding psychedelic and their use in clinical practice is in its infancy. A recent survey indicated that US psychiatrists (n=106) largely believed in the therapeutic potential of psychedelics and were strongly in favour of federal funding to facilitate further research with these substances.[23] Variables associated with a favourable assessment of psychedelics included greater psychedelic knowledge, fewer concerns about the drugs' addictive potential and working in research. A 2016 survey study led by the same author had also suggested that psychiatrists (n=324) were supportive of further psychedelic research on mental health disorders, but the sample also tended to view these drugs as potentially dangerous and agreed with their restrictive scheduling for recreational use.[24] A 7-year follow-up survey (n=131) found that optimism among psychiatrists regarding the clinical potential of psychedelics had increased while concerns about risk had decreased; 50% of psychiatrists indicated moderate or strong intentions to integrate psychedelic therapy into their clinical practice.[25] A pilot study (n=83) of psychiatrists working in the UK's National Health Service showed that most of them (77%) believed that controlled or therapeutic use of psychedelics should play a role in society.[26] Notably, psychiatrists of all grades reported significant training needs and lacking preparedness or confidence to deliver psychedelic-assisted psychotherapy should it become available. Overall, most research conducted in this nascent field converges on the conclusion that a majority of psychiatrists and other mental health professionals— including psychologists and counsellors[27 28]—are supportive of further clinical research with psychedelics.

Given that tomorrow's psychiatrists are today's medical students, it is pertinent to consider medical students' knowledge and attitudes towards psychedelics. To date, only three studies on this topic have been published.[29–31] In a sample of 213 US medical students, 83% and 52% of respondents believed that psilocybin should be legalised for recreational use and medical use, respectively.[29] Students with greater self-reported knowledge of psilocybin and fewer concerns about potential negative effects reported more positive overall perceptions regarding the medical use of psilocybin. Another survey study of US medical students (n=132) suggested that almost all respondents (95%) believed that further research examining psychedelics' therapeutic potential is warranted.[30] Overall, medical students held positive attitudes towards psychedelics, but only 41% reported that they felt knowledgeable about psychedelics. None of the demographic variables, including age, sex, planned specialty (i.e., psychiatry, internal medicine), and level of training, were associated with greater positive attitudes towards psychedelics. Lastly, another survey of US medical students (n=295) indicated that more favourable attitudes towards psychedelics were associated with a greater willingness to recommend psychedelic-assisted therapy to patients should it receive FDA (Food and Drug Administration) approval.[31] The present study aimed to further expand knowledge in this area by sampling the views and attitudes of medical students in the UK.

Using an anonymous online survey, we aimed to capture UK medical students' self-reported knowledge of psychedelics and their application in clinical research. We also aimed to compare medical students' harm assessment of psychoactive drugs with that of drug experts. We aimed to examine to which extent participant characteristics were associated with support for changing the legal status of psychedelics to facilitate further research. We expected more time spent in medical school to be associated with higher psychedelic knowledge and lower psychedelic harm composite scores. Regarding support for changing the legal status of psychedelics to facilitate further research, we expected students more knowledgeable of psychedelics to be more supportive of their legal status change; students assigning lower harm to psychedelics to be more supportive; more seasoned students to be more supportive; more psychiatrically interested students to be more supportive; students favourably assessing pharmacological and non-pharmacological to be more and less supportive, respectively; students more open to new experiences to be more supportive; and students with higher levels of neuroticism to be less supportive.

## METHODS
### Procedures and survey design

This study used an anonymous online survey delivered via the Qualtrics web platform to collect cross-sectional data without recording participants' IP addresses. A short text asking individuals whether they would like to support research on medical students' knowledge and perceptions of common 'recreational' drugs and their clinical uses was shared on UK university medical society mailing lists and bulletins, and social media platforms (LinkedIn, Twitter). To reduce self-selection bias, the study was advertised as a 'Drugs and Mental Health' survey, rather

than a survey about psychedelics. Interested individuals could follow a link to the study information sheet, which described the aims of the survey, the anonymous data collection, and the eligibility criteria (i.e., at least 18 years old, good understanding of the English language, currently enrolled in a UK medical programme accredited by the General Medical Council). Before starting the survey, individuals were required to indicate their informed consent by agreeing to the following statement: 'I have read and understand the explanations, I am at least 18 years old and have a good understanding of the English language, and I voluntarily consent to participate in this study'. The survey took approximately 10 min to complete and was accessible between October 2021 and March 2022. No financial or other incentives were offered for participation. All data were collected and processed in accordance with the 2018 UK General Data Protection Regulation. The full survey can be found in online supplemental material.

## Participants

A total of 136 individuals consented to participate and completed the survey. Incomplete responses were automatically deleted by Qualtrics after 1 month. In the service of survey quality control, participants who took less than 5 min to complete the survey were excluded, because this presents the time it approximately took to read the survey questions at a fast pace, but without responding to any questions (i.e., taking less than 5 min was judged to be indicative of participants not properly reading the survey questions or response options). Four participants were thus excluded. The total sample consisted of medical 132 students without missing data.

## MEASURES

To ensure anonymity, we collected only minimal demographic information including course year (i.e., years since starting training), participation in intercalated BSc or MBPhD programme, previous completion of university degree and interest in pursuing a career in psychiatry.

To capture medical students' self-reported knowledge and the amount of education they had received on seven drugs (i.e., cannabis, cocaine, heroin, ketamine, LSD, magic mushrooms/psilocybin, ecstasy/MDMA), we used the following three statements:

1. Self-reported knowledge: 'First, please indicate how well educated/knowledgeable you feel about these drugs. For example, this would include knowledge of pharmacology, physiology, subjective effects and adverse consequences of the drug.'
2. Encountered in curriculum: 'How often have you come across the drugs listed above in your teaching, including lectures, small group work and problem-based or self-directed learning?'
3. Encountered in extracurricular reading: 'How often have you come across these drugs in your own reading/research?'

The first statement used a sliding Visual Analogue Scale (VAS) ranging from 0 ('no knowledge') to 100 ('excellent knowledge'), while the latter two questions used a VAS from 0 ('never') to 100 ('very often'). Commonly used non-psychedelic drugs were included to allow comparison of relative perceived harms, as well as to cover the fact that this was primarily a psychedelic survey. A psychedelic knowledge composite was computed by averaging the knowledge scores of LSD, psilocybin, MDMA and ketamine (i.e., four scale scores were combined), with higher composite scores indicating greater self-reported knowledge.

To measure medical students' perceptions of harms of the same seven drugs, the following six questions with a VAS from 0 (no harm) to 100 (severe harm) were used:

4. Drug-specific mortality: 'How great is the potential of the drug to cause death by overdose (i.e., the intrinsic lethality)?'
5. Drug-related mortality: 'To what extent is life shortened by the use of the drug, excluding deaths from overdose? For example, this could include deaths due to road traffic accidents, lung cancers, HIV and suicide.'
6. Drug-specific damage: 'How great is the damage to physical health (short of death) caused by the drug? For example, this could include liver damage, other organ damage, seizures, strokes.'
7. Dependence: 'To what extent does the drug cause addiction/dependence, that is, the urge to continue to use despite adverse consequences?'
8. Drug-specific impairment of mental functioning: 'To what extent does the drug cause acute mental health problems when taken?'
9. Drug-related impairment of mental functioning: 'To what extent does the drug cause problems in the user's work or social life, leading to mental health problems (excluding addiction)?'

A psychedelic harm composite score was computed by averaging the harm scores of LSD, psilocybin, MDMA and ketamine across the six harm assessment categories (i.e., 24 scale scores were combined), with higher composite scores indicating higher levels of harm. The harm-related questions were adopted from a cross-sectional survey[32] that used the 16 harm criteria introduced by a multidisciplinary panel of drug experts in a seminal paper on relative drug harms on behalf of the in the UK's Independent Scientific Committee on Drugs.[33] Using a brief survey to minimise attrition rate, we adopted only 6 of the 16 criteria covering physical harm (drug-specific mortality, drug-related mortality, drug-specific damage) and psychological harm (dependence, drug-specific impairment of mental functioning, drug-related impairment of mental functioning). Detailed information about these harm criteria can be found in Nutt et al.[33]

To capture medical students' attitudes towards mental health treatments and research on psychedelics, we used the following questions and scales:

10. Perception of effectiveness of pharmacological (e.g., selective-serotonin reuptake inhibitors and

**Table 1** Demographics and participant characteristics (N=132)

| Variable | Mean (SD) or n (%) |
|---|---|
| Course years (i.e., time spent in medical education) | 3.8 (1.4), range: 1–6 |
| First year | 6 (4.6%) |
| Second year | 14 (10.6%) |
| Third year | 40 (30.3%) |
| Fourth year | 32 (24.2%) |
| Fifth year | 17 (12.9%) |
| Sixth year | 23 (17.4%) |
| Enrolled in or completed an intercalated degree | 62 (47.0%) |
| Attained a previous university degree | 29 (22.0%) |
| Interest in psychiatry | |
| I am confident I will pursue a career in psychiatry. | 8 (6.1%) |
| I consider psychiatry one of my more likely career paths. | 35 (26.5%) |
| I have not ruled out psychiatry, but it is not one of my more likely career paths. | 39 (29.6%) |
| I consider psychiatry one of my less likely career paths. | 35 (26.5%) |
| I am confident I will not pursue a career in psychiatry. | 15 (11.4%) |
| Influence on perceptions of drugs (range: 0 (not important) to 100 (very important)) | |
| My own and my friends' personal experiences | 77 (20) |
| Teaching as part of my course | 44 (29) |
| Media reporting | 39 (26) |
| Reading scientific papers/articles | 65 (28) |
| A drug's legal class | 28 (29) |
| Advice from parents or other authority figures | 34 (28) |
| General internet searches and online information | 61 (24) |
| Awareness of and interest in psychedelic research | |
| I am aware of this, and am well educated on the topic. | 22 (16.7%) |
| I am aware of this, and curious to know more. | 84 (63.6%) |
| I am aware of this, but it doesn't interest me. | 3 (2.3%) |
| I am not aware of this, but am curious to know more. | 22 (16.7%) |
| I am not aware of this, and it doesn't interest me. | 1 (0.8%) |
| Perception of mental health treatments (0 (not effective) to 100 (extremely effective)) | |
| Pharmacological (e.g., selective-serotonin reuptake inhibitors) | 58.7 (19.7) |
| Psychotherapy | 66.9 (20.9) |
| Openness to experience (BFI-10) | 3.72 (0.92) |
| Conscientiousness (BFI-10) | 3.76 (0.88) |
| Extraversion (BFI-10) | 3.61 (0.98) |
| Agreeableness (BFI-10) | 3.63 (0.86) |
| Neuroticism (BFI-10) | 3.01 (1.14) |

BFI-10, 10-item Big Five Inventory.

non-pharmacological treatments (e.g., psychotherapy): 'Thinking about mental health conditions such as anxiety disorders, depression and addiction, how effective would you consider the following general treatment approaches for these conditions?' (VAS from 0 (not effective at all) to 100 (extremely effective))

11. Awareness of and interest in psychedelic research (adapted from the Psychedelic as Medicine Report[34]): 'Consider the following statement: 'Drugs such as psilocybin (the active ingredient in magic mushrooms) and MDMA (the active ingredient in ecstasy) have recently been investigated as potential treatments for mental health conditions such as depression, PTSD

**Table 2** Descriptive statistics of medical students' self-reported knowledge and frequency of encountering psychedelics in teaching and in extracurricular reading (N=132)

|  | Self-reported knowledge* | Encountered in curriculum† | Encountered in extracurricular reading† |
|---|---|---|---|
| Cannabis | 75.6 (17.7) | 43.9 (28.4) | 60.1 (26.5) |
| Cocaine | 65.6 (22.2) | 42.5 (27.5) | 46.0 (28.4) |
| Heroin | 60.2 (21.7) | 39.5 (29.8) | 41.3 (30.0) |
| Ketamine | 61.2 (22.5) | 32.2 (27.5) | 41.9 (28.9) |
| LSD | 56.1 (27.9) | 16.8 (19.7) | 39.8 (33.7) |
| Magic mushrooms | 57.5 (29.7) | 14.1 (19.9) | 36.4 (33.0) |
| MDMA (ecstasy) | 64.5 (23.4) | 27.8 (25.5) | 40.0 (31.1) |

All variables represent the mean (and SD).
*Ranging from 0 (no knowledge) to 100 (excellent knowledge).
†Ranging from 0 (never) to 100 (very often).
LSD, lysergic acid diethylamide; MDMA, 3,4-methylenedioxymethamphetamine.

and addiction.' Which of the following best describes your own awareness of this research area?' (5-point nominal scale: I am aware of this, and am well educated on the topic, I am aware of this, and curious to know more; I am aware of this, but it doesn't interest me; I am not aware of this, but am curious to know more; I am not aware of this, and it doesn't interest me).

12. Attitude towards changing legal status of psilocybin and MDMA: 'Consider the following statement: 'Drugs that are considered to have no medical value and a high risk of misuse or harm are listed in Schedule 1 of the UK Misuse of Drugs Regulations (2001); these currently include MDMA and psilocybin. Research involving schedule 1 drugs is associated with significantly increased costs, duration and difficulty.' Given your current knowledge, to what extent would you support or oppose changing the legal status of these drugs to facilitate further research into their potential medical uses (while keeping restrictions on recreational use as they are)?' (VAS from 0 (strongly oppose) to 100 (strongly support))

A 'legal change' composite variable was created by averaging the psilocybin and MDMA legal change scores, with higher scores indicating greater support for changing the legal status to facilitate further research.

Another drug-related question asked medical students to rate, using a VAS from 0 (not important) to 100 (very important), the importance of the following factors in influencing their perception of the drugs included in this survey: my own and my friends' personal experiences; teaching as part of my course; media reporting; reading scientific papers/articles; a drug's legal class; advice from parents or other authority figures; general internet searches and online information.

To measure personality traits, the short version of the Big Five Inventory (BFI) was used.[35] This 10-item scale uses a 5-point Likert scale ranging from 1 (disagree strongly) to 5 (agree strongly) to capture openness to experience, conscientiousness, extraversion, agreeableness and neuroticism.[36] Each personality trait is captured by two items and total scores for each trait are computed by averaging the respective items scores. Higher subscale scores relate to higher trait levels. The 10-item version, despite being significantly shorter than the standard 44-item BFI on which it is based, has retained good levels of reliability and validity.[35]

### Data analysis
#### Sample size
Given the lack of studies that have used regression modelling to investigate the potential predictors of medical students' support for changing the legal status of psychedelics, we were not able to conduct an a priori power analysis based on a published effect size of this relationship. Instead, we aimed to recruit as many UK medical students as possible over a 6-month period.

#### Associations of drug-related measures with course year
The relationship between course year and drug-related measures was assessed using Pearson's correlation coefficients.

#### Regression analysis
We built the statistical models within a risk prediction framework using a stepwise backward elimination approach. The full mode considered the legal change composite score as the continuous outcome. Given the lack of research on medical students' attitudes towards psychedelic research that could guide the selection of potential predictors, we decided to include the following variables as continuous explanatory variables: psychedelic knowledge composite scores, psychedelic harm composite scores, course year, interest in psychiatry, perception of effectiveness of mental health treatments, openness to experience and neuroticism. The explanatory variable associated with the highest p value above 0.05 was then deleted. The model was then refitted without the deleted explanatory variable and the estimates and their associated p values recomputed. The final prediction model

**Table 3** Multidimensional harm assessment of seven drugs (N=132)

| | Drug-specific mortality | Drug-related mortality | Drug-specific damage | Dependence | Drug-specific impairment of mental functioning | Drug-related impairment of mental functioning |
|---|---|---|---|---|---|---|
| Cannabis | 13.3 (17.6) | 38.2 (25.5) | 34.9 (25.5) | 48.5 (27.5) | 58.4 (27.4) | 48.9 (26.7) |
| Cocaine | 70.8 (21.4) | 66.4 (20.2) | 73.5 (17.9) | 80 (16.9) | 64 (23.7) | 68.4 (22.8) |
| Heroin | 89.9 (14.3) | 85.3 (16.9) | 84.5 (15.7) | 93.9 (11.9) | 69.5 (25.5) | 89.3 (13.8) |
| Ketamine | 57.4 (27.7) | 49.4 (25) | 54.6 (24.8) | 49.7 (25.8) | 52.6 (25) | 49.6 (24.9) |
| LSD | 35.5 (28.7) | 39.4 (28.1) | 36.5 (28.9) | 31.1 (26.5) | 60.9 (26.1) | 39.4 (27.3) |
| Magic mushrooms | 28.1 (26.2) | 32.3 (25.6) | 29.4 (24.6) | 27.2 (23.7) | 55.2 (28) | 36.5 (27.2) |
| MDMA (ecstasy) | 63.5 (26.7) | 53.7 (25.3) | 55.9 (25.6) | 48.1 (26.6) | 59 (26.2) | 48 (24.6) |

All values represent the mean (and 95% CI) on a scale ranging from 0 (no harm) to 100 (severe harm).
LSD, lysergic acid diethylamide; MDMA, 3,4-Methylenedioxymethamphetamine.

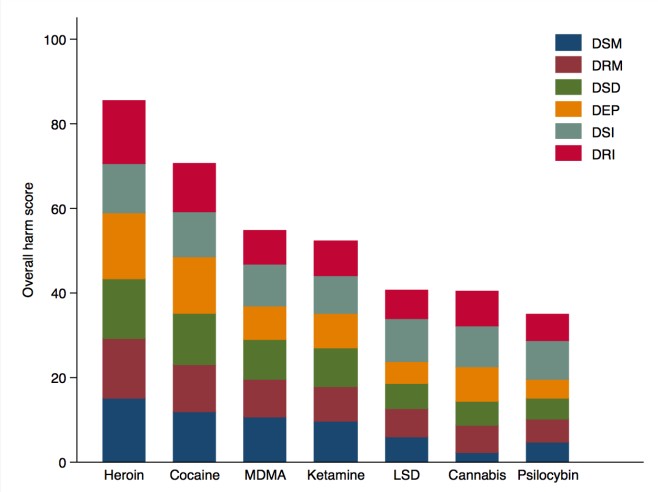

**Figure 1** Drugs ordered by their overall harm scores, showing subscores for each of the six harm criteria. Harm scores range from 0 (no harm) to 100 (severe harm). DEP, dependence; DRI, drug-related impairment; DRM, drug-related mortality; DSD, drug-specific damage; DSI, drug-specific impairment; DSM, drug-specific mortality; LSD, lysergic acid diethylamide; MDMA, 3,4-methylenedioxymethamphetamine.

thus retained explanatory variables if they were associated with p<0.05. This threshold for inclusion was chosen to ensure model stability and reduce overfitting. Stata/MP V.16.0 and R V.4.0.2 were used for all analyses.

### Patient and public involvement

Patients or the public were not involved in the development, design, conduct, reporting or dissemination plans of our research.

### RESULTS
### Descriptive statistics

Participant characteristics for the total sample of 132 medical students are displayed in table 1. We used Qualtrics survey settings that deleted incomplete responses after 1 month. A response rate could, therefore, not be computed. Descriptive statistics of self-reported knowledge and the extent to which medical students had encountered a discussion of the listed drugs (i.e., cannabis, cocaine, heroin, ketamine, LSD, psilocybin, MDMA) in their curriculum or during extracurricular reading and research are displayed in table 2. Descriptive statistics on medical students' assessment of the harm caused by the listed drugs are reported in table 3 and visually depicted in figure 1.

The mean rating (on a scale from 0 (strongly oppose) to 100 (strongly support)) for the change in legal status for psilocybin was 80.2 (SD=24.8) and for MDMA was 74.3 (SD=27.7). Both distributions were left-skewed (i.e., more values are on the right side of the mean) with 59 (44.7%) and 50 (37.9%) medical students having a score of 100 for psilocybin and MDMA, respectively. The legal

**Table 4** Multivariable linear regression models of the association with support for legal change (N=132)

| | Legal change composite scores* | | |
| | Full model | | |
| **Explanatory variables (unstandardised)** | **Estimate** | **95% CI** | **P value** |
|---|---|---|---|
| Psychedelic knowledge | 0.31 | 0.13 to 0.48 | 0.001 |
| Psychedelic harm | −0.43 | −0.66 to −0.21 | <0.001 |
| Course year (1 year) | 2.82 | 0.16 to 5.48 | 0.038 |
| Interest in psychiatry | −0.68 | −4.01 to 2.65 | 0.687 |
| Perception of pharmacological treatments | −0.01 | −0.22 to 0.2 | 0.927 |
| Perception of non-pharmacological treatments | −0.19 | −0.38 to 0 | 0.051 |
| Openness to experience | 1.62 | −2.4 to 5.64 | 0.426 |
| Neuroticism | −0.10 | −3.35 to 3.16 | 0.952 |
| (Intercept) | 77.53 | 48.43 to 106.64 | <0.001 |
| | **Final prediction model†** | | |
| **Explanatory variables (unstandardised)** | **Estimate** | **95% CI** | **P value** |
| Psychedelic knowledge | 0.31 | 0.14 to 0.48 | <0.001 |
| Psychedelic harm | −0.45 | −0.65 to −0.24 | <0.001 |
| Course year (1 year) | 3.00 | 0.4 to 5.59 | 0.024 |
| Perception of non-pharmacological treatments | −0.18 | −0.36 to 0 | 0.044 |
| (Intercept) | 79.83 | 59.26 to 100.40 | <0.001 |

*Ranging from 0 (strongly oppose) to 100 (strongly support).
†The final model retained all explanatory variables with p<0.05.

change composite score showed a similar divergence from normality (skewness: −0.98, kurtosis: 3.02).

The other two composite scores did not markedly diverge from normality as indicated by estimates of skewness (psychedelic knowledge: −0.17; psychedelic harm: −0.13) and kurtosis (psychedelic knowledge: 2.30; psychedelic harm: 2.34) and visual inspection of the histograms.

### Associations of drug-related measures with course year
More course years were associated with lower psychedelic harm composite scores (Pearson's correlation coefficient r=−0.22, 95% CI −0.05 to −0.38, p=0.012). In contrast to our expectations, a number of course years were not correlated with higher psychedelic knowledge composite scores (r=0.12, 95% CI −0.04 to 0.30, p=0.129).

### Regression analysis
Table 4 reports the results from the full model and the final prediction model. Using a stepwise backward elimination approach with significance threshold of p=0.05, explanatory variables were removed in the following order: neuroticism, perception of pharmacological treatments, interest in psychiatry, and, in the last step, openness to experience. The final prediction model for legal change composite scores thus retained psychedelic knowledge composite scores, psychedelic harm composite scores, course year and perception of the effectiveness of non-pharmacological treatments (all unstandardised estimates are displayed in table 4). In other words, in the final regression model, greater psychedelic

knowledge and more course years were associated with greater support for legal change, whereas greater psychedelic harm scores and a more favourable perception of the effectiveness of non-pharmacological treatments were associated with weaker support for legal change of psychedelics. Assumptions of linear regression—namely, independence, normality (all W-values of Shapiro-Wilk test for normal data associated with p>0.123), and homoscedasticity of residuals and linearity—were judged to be adequately met for all regression models; no notable level of multicollinearity was detected.

### DISCUSSION
The primary aim of this cross-sectional study was to capture UK medical students' self-reported knowledge and harm assessment of psychedelics and the degree to which they would support changing the legal status of psychedelic substances, particularly psilocybin and MDMA, to facilitate further research into their potential medical uses without changing restrictions on recreational use. Results indicated that 83% of medical students were aware of psychedelic research and only four participants said that they were not interested in learning more about this type of research. Medical students evaluated their own experience and that of their friends as the most important influence on their perception of drugs in general. Another important influence was their study of scientific articles,

whereas media reporting and a drug's legal class had the least influence on their perception of drugs.

UK medical students were highly supportive of changing the legal status of psilocybin and MDMA, with 45% and 38% expressing the strongest possible support (i.e., a score of 100 on a scale from 0 (strongly oppose) to 100 (strongly support)) for psilocybin and MDMA, respectively. Relatedly, we aimed to explore the variables that would predict the extent to which medical students were in support of changing the legal status. Regression modelling suggested that students who reported greater knowledge of psychedelics, assigned lower harm scores to psychedelics, had spent more time spent in medical education and assigned lower effectiveness scores to non-pharmacological mental health treatments showed a greater support for legal status change. The contribution of these variables was in line with our prediction. However, diverging from our expectation was the absence of a relationship between support for legal change and an interest in psychiatry, the perceived effectiveness of pharmacological treatments, and the personality traits of openness to experience and neuroticism.

Although time spent in medical school itself was associated with lower harm ratings of psychedelics, it was not associated with greater self-reported knowledge of psychedelics, suggesting that medical degree curricula do not contain much teaching about psychedelics. This explanation also aligns with the low reported frequency with which medical students encountered the classic psychedelics, psilocybin and LSD, in their curriculum. An alternative explanation, for instance, that medical students attracted to a survey on drugs and mental health tend to have greater knowledge of psychedelics (resulting in ceiling effects), was not reflected in the data, which showed self-reported psychedelic knowledge to be approximately normally distributed. In this context, it is important to consider that self-treatment with psychedelics is increasing[37 38] and that harm reduction opportunities for psychedelic use among patients could be potentially missed due to lack of knowledge about psychedelics among healthcare providers. Specifically, patients are afraid to discuss psychedelic use with providers due to concerns about stigma and the assumption that their providers' knowledge base is low.[39] For example, providers could encourage patients with bipolar or psychotic disorders to avoid psychedelics and inform them about potentially adverse medication interactions such as seizures from combining psychedelics with lithium.[40]

Medical students' multidimensional harm assessment of seven drugs resulted in an order of assigned overall harm scores that closely matched the order that was assigned by drug expert in a widely-cited harm assessment report conducted on behalf of the UK's Independent Scientific Committee on Drugs.[33] In alignment with the expert rating, medical students assigned the highest overall harm score to heroin and cocaine and the lowest to psilocybin. Differences emerged in relation to the rank order of cannabis, which experts rated as substantially more harmful than medical students. In contrast, medical students perceived MDMA to be the third most harmful substance among the list of seven drugs assessed in the present study, whereas experts judged the harm of MDMA to be similar to that of LSD and psilocybin.

## Limitations

This study has important limitations that need to be considered when interpreting our results. First, the cross-sectional nature of our data prevents us from drawing causal conclusions about the association between support for changing the legal status of psychedelics and the potential predictors we have considered. Future research with larger sample sizes, more detailed demographic information, objective assessments of students' knowledge, and longitudinal follow-ups are needed to further understand how medical education shapes students' perception of psychedelics and the factors that influence the degree to which students' views align with empirical evidence and expert opinion. Second, some of the measures we applied to capture medical students' knowledge and attitudes towards psychedelics were tailor-made for this survey and not previously validated, which needs to be considered when interpreting the results. Our harm assessment was adapted from Nutt et al,[33] but we used only 6 of the 16 harm criteria these researchers proposed; and none of our harm criteria captured harm to others. We also did not include an assessment of other commonly used drugs, such as alcohol and tobacco, preventing us from drawing potentially insightful comparisons. The brevity of our assessment was to keep attrition rates as low as possible. We did not capture medical students' personal psychedelic use, which may be associated with attitudes towards psychedelics.[41] Future research would benefit from a more in-depth and comprehensive assessment of UK medical students' views and attitudes surrounding psychedelics and their application in clinical research. Third, we relied on self-report data gathered via an anonymous online survey, which automatically deleted incomplete responses after 1 month, thus not allowing us to capture a response rate; we did not conduct an objective assessment of knowledge about psychedelics (e.g., a multiple choice test) and, therefore, cannot assess the validity and reliability of students' self-report. Lastly, generalisability of our study may be limited due to the present sample's high awareness of and interest in psychedelic research. Despite efforts to minimise selection bias by avoiding the term 'psychedelic' during recruitment, it is likely that the students who completed the survey were not representative of the general UK medical student population. The generalisability of our findings might also be impeded by the heterogeneity of the sample in relation to length of medical training. Overall, given the lack of empirical and theoretical work in this area, our hypotheses and statistical approaches should be viewed as primarily exploratory and our findings as preliminary (our data are openly accessible).[42]

## CONCLUSION

This is the first study to explore UK medical students' self-reported knowledge and perceptions of psychedelics. Medical students showed strong support for changing the legal status of psychedelics to facilitate further research into their potential clinical applications. In general, medical students only rarely encountered teachings on psychedelics in their medical education. While only a small percentage of medical students felt well-educated on psychedelic research, a large majority were interested to learn more, likely reflecting the re-emergence of clinical interest in this area.

**Author affiliations**
[1]UCL Medical School, University College London, London, UK
[2]Department of Psychiatry, University of Oxford, Oxford, UK
[3]Wellcome Centre for Ethics and Humanities, University of Oxford, Oxford, UK
[4]The Department of Psychological Medicine, King's College London, London, UK
[5]South London and Maudsley NHS Foundation Trust, Kent, UK
[6]University of Exeter Medical School, Exeter, UK
[7]Innerspace Institute, Alentejo, Portugal
[8]Division of Psychiatry, UCL, London, UK
[9]Institut für Psychotherapie Potsdam, Potsdam, Germany

**Contributors** CS-S: conception of the work, methodology, data acquisition, data analysis, reviewing the work critically for important intellectual content. EJ: conception of the work, methodology, reviewing the work critically for important intellectual content, supervision. JR: conception of the work, reviewing the work critically for important intellectual content. MS: conception of the work, reviewing the work critically for important intellectual content. JC: conception of the work, methodology, reviewing the work critically for important intellectual content, supervision. MS: writing of the original draft, conception of the work, methodology, data analysis, data interpretation, visualisation, supervision. MS is responsible for the overall content as the guarantor.

**Funding** This research was funded in whole, or in part, by the Wellcome Trust 221152/Z/20/Z. For the purpose of Open Access, the author has applied a CC BY public copyright licence to any Author Accepted Manuscript version arising from this submission. JR received salary support via the NIHR Clinician Scientist Fellowship Programme during the period of this research (CS-2017-17-007).

**Competing interests** JR has received payments for advisory work from Clerkenwell Health (Past), Beckley PsyTech (Past), Delica Therapeutics (Past), for articles written for Janssen, and financial assistance for attendance at conferences from Compass Pathways (past) and Janssen. JR has received grant funding (received and managed by King's College London) from Compass Pathfinder, Beckley PsyTech, Multidisciplinary Association for Psychedelic Studies, National Institute for Health Research, Wellcome Trust, Biomedical Research Centre at the South London and Maudsley NHS Foundation Trust. All other authors declare that they have no competing interests.

**Patient and public involvement** Patients and/or the public were not involved in the design, or conduct, or reporting, or dissemination plans of this research.

**Patient consent for publication** Not applicable.

**Ethics approval** This study received ethical approval from University College London's Research Ethics Committee (Project ID: 14309/007) and was performed in accordance with the 1964 Declaration of Helsinki and its later amendments.

**Provenance and peer review** Not commissioned; externally peer reviewed.

**Data availability statement** Data are available in a public, open access repository. All data and analysis scripts can be found on the senior author's Open Science Framework profile: https://osf.io/tmp3u/.

**ORCID iDs**
James Rucker http://orcid.org/0000-0003-4647-8088
Marco Schlosser http://orcid.org/0000-0002-0634-0749

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
