## [Reviewer comments · BMJ Open]

ARTICLE DETAILS

TITLE (PROVISIONAL)	UK medical students' self-reported knowledge and harm assessment of psychedelics and their application in clinical research: a cross-sectional study
AUTHORS	Song-Smith, Charlie; Jacobs, Edward; Rucker, James; Saint, Matthew; Cooke, James; Schlosser, Marco

VERSION 1 – REVIEW

REVIEWER	Buljan, Ivan Sveuciliste u Splitu Medicinski fakultet
REVIEW RETURNED	24-Jan-2024

GENERAL COMMENTS	Thank you for the opportunity to review this manuscript. The authors have conducted the study about attitudes towards psychedelics in small sample of UK medical students. The results suggest that the attitudes were very positive, and students indicate that they would benefit from additional education about psychedelics. My principal concern is about study power. You need to elaborate that your study is powered enough to make conclusions you stated in the article. Also, it is unclear what was response rate in the study, and finally there was no validated measure for measurement of attitudes towards psychedelics. All of this must be stated in the methods and limitations section, and it may affect interpretation. Some of the recent references in assessment of psychedelics are missing, especially in assessment of attitudes towards psychedelics. It is not clear how students gave informed consent, please provide the document in the appendix. For every outcome measure, please state what were the answer options. Also, I suggest that you itemize Methods according to STROBE guidelines. What is the justification for exclusion of participants who took less than 5 minutes, who set that criterion? For data availability, please include results along the code, not only data. For distribution testing, please provide P values. Please list (at least in the appendix) all assumptions which were met for regression analysis, and include intercept in the tables. The students were very heterogeneous by years of study. Address this generalization problem in the discussion
--

REVIEWER	Barnett, Brian Cleveland Clinic, Department of Psychiatry and Psychology
REVIEW RETURNED	24-Jan-2024

GENERAL COMMENTS	This is a survey study investigating an important topic related to medical student exposures to psychedelics in their learning and their attitudes towards them. It is well designed, and I have only minor suggestions to improve it.  • Intro  o At beginning of paragraph one, I would make the context a little bit clearer by stating that several companies are undertaking efforts to commercialize psychedelics as medical treatments. I recommend citing this paper: Psychedelic Commercialization: A Wide-Spanning Overview of the Emerging Psychedelic Industry. https://www.liebertpub.com/doi/full/10.1089/psymed.2023.0013 o I would also mention that in addition to prohibitive legislation, a lack of governmental research funding (https://pubmed.ncbi.nlm.nih.gov/34624734/) played an important role in halting this research. o Later in that paragraph, I would cite a recent survey showing a significant shift in pro-psychedelic attitudes among US psychiatrists: American Psychiatrists' Opinions About Classic Hallucinogens and Their Potential Therapeutic Applications: A 7-Year Follow-Up Survey https://www.liebertpub.com/doi/full/10.1089/psymed.2023.0036 • Methods  o If participants were told to define psychedelic in a certain way, please include that (varying definitions could exclude or include MDMA, ketamine, PCP, marijuana, etc) • Discussion  o I would amplify the importance of the findings a bit by discussing the fact that self treatment with psychedelics is increasing (https://pubmed.ncbi.nlm.nih.gov/36876583/; https://pubmed.ncbi.nlm.nih.gov/37819655/) and that harm reduction opportunities for psychedelic use among patients are being missed due to lack of knowledge about psychedelics among healthcare providers (this survey confirms little didactic exposure to this topic). Specifically, patients are afraid to discuss psychedelic use with providers due to worries about stigma and the assumption that their providers knowledge base is low (https://pubmed.ncbi.nlm.nih.gov/37599880/). For example, providers could tell patients with bipolar or psychosis to avoid psychedelics (can cite some relative papers on this adverse event) and about the possibility of medication interactions such as seizures from combining psychedelics with lithium (Classic Psychedelic Coadministration with Lithium, but Not Lamotrigine, is Associated with Seizures: An Analysis of Online Psychedelic Experience Reports - PubMed (nih.gov)). • Limitations  o I would mention that you did not ask about personal use which may influence attitudes towards psychedelics. There are no data on this issue explicitly, but it might be useful to note that psychedelic use is high among psychedelic therapists who are obviously highly supportive of using psychedelics as treatments (https://www.liebertpub.com/doi/full/10.1089/psymed.2022.0004)
---

VERSION 1 – AUTHOR RESPONSE

Reviewer #1

My principal concern is about study power. You need to elaborate that your study is powered enough to make conclusions you stated in the article.

Our response: We thank the reviewer for sharing their concern regarding the study's statistical power. To clarify our approach, we have now added the following section to the methods section (page 12, third paragraph): "Sample size: Given the lack of studies that have utilised regression modelling to investigate the potential predictors of medical students' support for changing the legal status of psychedelics, we were not able to conduct an a priori power analysis based on a published effect size of this relationship. Instead, we aimed to recruit as many UK medical students as possible over a 6-month period. Given this context, our hypotheses and statistical approaches should be viewed as primarily exploratory and the findings as preliminary."

Also, it is unclear what was response rate in the study, and finally there was no validated measure for measurement of attitudes towards psychedelics. All of this must be stated in the methods and limitations section, and it may affect interpretation.

Our response: We thank the reviewer for encouraging us to specify these limitations in our discussion section. We have now added/modified the following sentences (page 17, second paragraph): "Second, some of the measures we applied to capture medical students' knowledge and attitudes towards psychedelics were tailor-made for this survey and not previously validated, which needs to be considered when interpreting the results." & Page 18, first paragraph: "Third, we relied on self-report data gathered via an anonymous online survey, which automatically deleted incomplete responses after one month, thus not allowing us to capture a response rate; we did not conduct an objective assessment of knowledge about psychedelics (e.g., a multiple choice test) and, therefore, cannot assess the validity and reliability of students' self-report."

Some of the recent references in assessment of psychedelics are missing, especially in assessment of attitudes towards psychedelics.

Our response: We have now added the relevant references based on the suggestions that were provided by reviewer 2.

It is not clear how students gave informed consent, please provide the document in the appendix.

Our response: The full questionnaire has now been uploaded as supplementary material. Further, the following sentence has been added (page 7, last paragraph): "Before starting the survey, individuals were required to indicate their informed consent by agreeing to the following statement: "I have read and understand the explanations, I am at least 18 years old and have a good understanding of the English language, and I voluntarily consent to participate in this study".

For every outcome measure, please state what were the answer options.

Our response: Response options for all measures are included in the methods section. Whenever several outcome measures are listed in bullet-point format, response options are specified either in the preceding or subsequent paragraph.

Also, I suggest that you itemize Methods according to STROBE guidelines.

Our response: Following the editor's suggestion, we completed the CROSS checklist instead.

What is the justification for exclusion of participants who took less than 5 minutes, who set that criterion?

Our response: We thank the reviewer for prompting us to clarify the relevant justification. The following sentence has now been added to the methods section (page 8, second paragraph): "In the service of survey quality control, participants who took less than five minutes to complete the survey were excluded, because this presented the time it approximately took to read the survey questions at a fast pace, but without responding to any questions (i.e., taking less than five minutes was judged to be indicative of participants not properly reading the survey questions or response options). Four participants were thus excluded."

For data availability, please include results along the code, not only data.

Our response: Following BMJ Open's Tier 2 data policy, we only provide code and data, which can be used to replicate the results.

For distribution testing, please provide P values.

Please list (at least in the appendix) all assumptions which were met for regression analysis, and include intercept in the tables.

Our response: We thank the reviewer for encouraging us to improve the presentation of our regression results. We have now included a statistical test of the distribution of the residuals alongside a list of all regression assumptions. The following sentence has been added (page 15, first paragraph): "Assumptions of linear regression – namely, independence, normality (all Ws of Shapiro Wilk test for normal data associated with $p > 0.123$), and homoscedasticity of residuals, and linearity – were judged to be adequately met for all regression models; no notable level of multicollinearity was detected." The intercepts have been added to Table 4.

The students were very heterogenous by years of study. Address this generalization problem in the discussion.

Our response: We have added this limitation to the discussion section (page 18, first paragraph): "The generalisability of our findings might also be impeded by the heterogeneity of the sample in relation to length of medical training."

Reviewer #2

Intro

At beginning of paragraph one, I would make the context a little bit clearer by stating that several companies are undertaking efforts to commercialize psychedelics as medical treatments. I recommend citing this paper: Psychedelic Commercialization: A Wide-Spanning Overview of the Emerging Psychedelic Industry. <https://www.liebertpub.com/doi/full/10.1089/psymed.2023.0013>

Our response: We thank the reviewer for this helpful suggestion. Accordingly, we have now added the following sentence (page 4, last paragraph): “Several companies are undertaking efforts to commercialize psychedelics as medical treatments (see Aday et al., 2023).”

I would also mention that in addition to prohibitive legislation, a lack of governmental research funding (<https://pubmed.ncbi.nlm.nih.gov/34624734/>) played an important role in halting this research.

Our response: The following sentence has been modified accordingly (page 4, first paragraph): “After a short-lived period of medical use of psychedelics by clinicians and researchers in the middle of the 20th century, psychedelic research was effectively ended by prohibitive legislature and a lack of governmental research funding (Rucker et al., 2018).”

Later in that paragraph, I would cite a recent survey showing a significant shift in pro-psychedelic attitudes among US psychiatrists: American Psychiatrists' Opinions About Classic Hallucinogens and Their Potential Therapeutic Applications: A 7-Year Follow-Up Survey
<https://www.liebertpub.com/doi/full/10.1089/psymed.2023.0036>

Our response: We added the following sentence (page 5, first paragraph): “A 7-year follow-up survey (n=131) found that optimism among psychiatrists regarding the clinical potential of psychedelics had increased while concerns about risk had decreased; 50% of psychiatrists indicated moderate or strong intentions to integrate psychedelic therapy into their clinical practice (Barnett et al., 2023).”

Methods

If participants were told to define psychedelic in a certain way, please include that (varying definitions could exclude or include MDMA, ketamine, PCP, marijuana, etc)

Our response: Participants were not given specific definitions of psychedelics. We have now uploaded the full survey as supplementary material.

Discussion

I would amplify the importance of the findings a bit by discussing the fact that self treatment with psychedelics is increasing (<https://pubmed.ncbi.nlm.nih.gov/36876583/>; <https://pubmed.ncbi.nlm.nih.gov/37819655/>) and that harm reduction opportunities for psychedelic use among patients are being missed due to lack of knowledge about psychedelics among healthcare providers (this survey confirms little didactic exposure to this topic). Specifically, patients are afraid to discuss psychedelic use with providers due to worries about stigma and the assumption that their providers knowledge base is low (<https://pubmed.ncbi.nlm.nih.gov/37599880/>). For example, providers could tell patients with bipolar or psychosis to avoid psychedelics (can cite some relative papers on this adverse event) and about the possibility of medication interactions such as seizures from combining psychedelics with lithium (Classic Psychedelic Coadministration with Lithium, but Not Lamotrigine, is Associated with Seizures: An Analysis of Online Psychedelic Experience Reports - PubMed (nih.gov)).

Our response: We thank the reviewer for providing us with this relevant suggestion. We have refined our discussion section accordingly (page 16, last paragraph): “In this context, it is important to consider that self-treatment with psychedelics is increasing (Kopra et al., 2023; Walsh et al., 2024) and that harm reduction opportunities for psychedelic use among patients could be potentially missed

due to lack of knowledge about psychedelics among healthcare providers. Specifically, patients are afraid to discuss psychedelic use with providers due to concerns about stigma and the assumption that their providers' knowledge base is low (Boehne et al., 2023). For example, providers could encourage patients with bipolar or psychotic disorders to avoid psychedelics and inform them about potentially adverse medication interactions such as seizures from combining psychedelics with lithium (Nayak et al., 2021)."

Limitations

I would mention that you did not ask about personal use which may influence attitudes towards psychedelics. There are no data on this issue explicitly, but it might be useful to note that psychedelic use is high among psychedelic therapists who are obviously highly supportive of using psychedelics as treatments (<https://www.liebertpub.com/doi/full/10.1089/psymed.2022.0004>)

Our response: We are grateful to the reviewer for noticing this important omission. We have now added this limitation to the discussion section (page 18, first paragraph): "We did not capture medical students' personal psychedelic use, which may be associated with attitudes towards psychedelics (see Aday et al., 2023)."

VERSION 2 – REVIEW

REVIEWER	Buljan, Ivan Sveuciliste u Splitu Medicinski fakultet
REVIEW RETURNED	16-Feb-2024

GENERAL COMMENTS	The authors have adressed all my comments and I have nothing to add.
--

REVIEWER	Barnett, Brian Cleveland Clinic, Department of Psychiatry and Psychology
REVIEW RETURNED	16-Feb-2024

GENERAL COMMENTS	The authors have addressed my recommendations and I recommend acceptance of the manuscript.
---